# Connecting Neural Models Latent Geometries with Relative Geodesic Representations

**Hanlin Yu**
Department of Computer Science
University of Helsinki
hanlin.yu@helsinki.fi

**Berfin Inal**
Faculty of Science
University of Amsterdam
s.berfininal@gmail.com

**Marco Fumero**
Department of Computer Science
Institute of Science and Technology Austria
marco.fumero@ist.ac.at

## Abstract

Neural models learn representations of high dimensional data that lie on low dimensional manifolds. Multiple factors, including stochasticities in the training process, may induce different representations, even when learning the same task on the same data. However, when there exist a latent structure shared between different representational spaces, it has been showed that is possible to model a transformation between them. In this work, we show how by leveraging the differential geometrical structure of latent spaces of neural models, it is possible to capture precisely the transformations between distinct latent spaces. We validate experimentally our method on autoencoder models and real pretrained foundational vision models across diverse architectures, initializations and tasks.

## 1   Introduction

Recent research reveals that neural models often develop similar internal representations when exposed to similar inputs, a phenomenon observed in both biological [24, 17] and artificial systems [26, 32, 19, 21]. Remarkably, even when models have different architectures, their internal representations can frequently be aligned through a simple transformation, for example linear [27, 25, 30]. This suggests a certain consistency in how NNs encode information, emphasizing the importance of studying these internal representations, and the transformations that relate them. One strategy to do this is to identify representations that are *invariant* to transformations between distinct models representational spaces. A simple and effective recipe to do this is the one of relative representations [32], where samples are represented as a function of a fixed set of latent representations. The similarity function employed is cosine similarity, hinting at the fact that representations across distinct models are subject to *angle preserving* transformations. However, the choice of similarity function should not be limited to capture invariances to one class of transformations. As shown in [8], other choices can be good as well, and there's not a clear best choice among simple class of transformations for capturing transformation across distinct latent spaces.

In this paper we employ geodesic distance in the latent space as a metric for relative representations. This approach ensures that the relative space remains approximately invariant to the isometries of the data's manifold, as characterized by a Riemannian structure. Our contributions can be summarized as follows: (i) We propose a new representation that capture the isometric transformation between data manifolds learned by distinct models. (ii) We propose to employ an approximation of the underlying metric both for classification and reconstruction tasks. (iii) We test relative geodesics on retrieval and stitching tasks on autoencoders and real vision foundation models, across different seeds, architectures and training strategies, outperforming previous methods.

## 2 Method

### 2.1 Notation and Background

Neural networks (NNs) can be viewed as parametric functions $F_\theta$, which are composed of an *encoding* map and a *decoding* map, represented as $F_\theta = D_{\theta_2} \circ E_{\theta_1}$. The encoder $E_{\theta_1} : \mathcal{X} \mapsto \mathcal{Z}$ generates a latent representation $z = E_{\theta_1}(x)$, where $x \in \mathcal{X}$ to the input domain $\mathcal{X}$, and the latent space $\mathcal{Z}$. The decoder $D_{\theta_2}$ is responsible for performing the task at hand, such as reconstruction or classification. For simplicity, we will omit the parameter dependence ($\theta$) in our notation moving forward. For any single module $E$ (or equivalently $D$), we will use $E_\mathcal{X}$ to denote that the module $E$ was trained on the domain $\mathcal{X}$. In the next sections, we will provide the necessary background to introduce our method.

**Latent Space Communication** Given a pair of domains $\mathcal{X}, \mathcal{Y}$, a pair of neural models trained on them $F_\mathcal{X}^1, F_\mathcal{Y}^2$, and a partial correspondence between the domains $\Gamma : \mathcal{A}_\mathcal{X} \mapsto \mathcal{A}_\mathcal{Y}$ where $\mathcal{A}_\mathcal{X} \subset \mathcal{X}$ and $\mathcal{A}_\mathcal{Y} \subset \mathcal{Y}$, the problem of *latent space communication* is the one of finding a full correspondence $\Lambda : E^1(\mathcal{X}) \mapsto E^2(\mathcal{Y})$ between the two domains, from $\Gamma$. In a simplified setting, e.g. two models trained with different initialization or architectures on the same data $\mathcal{X} = \mathcal{Y}$ and the correspondence is the identity. When $\mathcal{X} \neq \mathcal{Y}$ the problem becomes multimodal.

**Relative representations** The relative representations framework [32] provides a straightforward approach to represent each sample in the latent space according to its similarity to a set of fixed training samples, denoted as *anchors*. Representing samples in the latent space as a function of the anchors corresponds to transitioning from an absolute coordinate frame into a *relative* one defined by the anchors and the similarity function. Given a domain $\mathcal{X}$, an encoding function $E_\mathcal{X} : \mathcal{X} \to \mathcal{Z}$, a set of anchors $\mathcal{A}_\mathcal{X} \subset \mathcal{X}$, and a similarity or distance function $d : \mathcal{Z} \times \mathcal{Z} \to \mathbb{R}$, the *relative representation* for a sample $x \in \mathcal{X}$ is:

$$RR(z; \mathcal{A}_\mathcal{X}, d) = \bigoplus_{a_i \in \mathcal{A}_\mathcal{X}} d(z, E_\mathcal{X}(a_i)),$$

where $z = E_\mathcal{X}(x)$, and $\bigoplus$ denotes row-wise concatenation. In [32], $d$ was set as cosine similarity. This choice induces a representation invariant to *angle-preserving transformations*. In this work, our focus is to *leverage the intrinsic geometry of latent spaces to capture isometric transformations between data manifolds approximations.*

**Latent space geometry** For the latent space of a neural network, it is in general hard to reason about its Riemannian structure. However, it is often easier to assign a Riemannian structure to the output space. As such, one can define a *pullback metric* from the output space to the latent space, which is a standard operation in Riemannian geometry (see e.g. Ch.2.4 of [12]).

Formally, considering the decoder $D : \mathcal{Z} \mapsto \mathcal{X}$ takes as input a latent representations $z \in \mathcal{Z}$ and outputs $x$. Given a Riemannian metric defined on $x$ as $G_\mathcal{X}(x)$. Then, the Riemannian metric at $z$ can be obtained as

$$G_\mathcal{Z}(z) = \left(\frac{\partial x}{\partial z}\right)^\top G_\mathcal{X}(x) \frac{\partial x}{\partial z} = J_z(D)^T J_z(D),$$

where $J_z(D)$ is the Jacobian of $D$ evaluated at $z$. The metric tensor $G_\mathcal{X}$ is useful to compute quantities such lengths, angles and areas on $\mathcal{M}$. Given a smooth curve $\gamma : [a, b] \mapsto \mathcal{M}$ one can compute the energy $\mathcal{E}$ of $\gamma$ as follows [34]

$$\mathcal{E}(\gamma) = \frac{1}{2} \int_a^b v(t)^\top G(t) v(t)^\top \mathrm{d}t, \tag{1}$$

where $v(t) = \dot{\gamma}(t)$. This can be approximated using finite difference approaches. Geodesics are minimizers of this energy [34].

### 2.2 Relative geodesics representations

When considering a differential geometry perspective, the problem of latent space communication can be interpreted as finding a transformation between the data manifolds $\mathcal{M}_1, \mathcal{M}_2$ approximated by two neural models $F_1, F_2$. The relative representation framework can capture this transformation implicitly if equipped with the right metric. A natural candidate for this metric is the geodesic distance defined on $\mathcal{M}_1, \mathcal{M}_2$, respectively. This choice make the relative representations invariant to isometric transformation of the manifolds $\mathcal{M}_1, \mathcal{M}_2$. However, for high dimensional problems, the high cost of computing the geodesic renders the above methods inappropriate [34, 9]. Furthermore, one can argue against directly using the latent

geometry induced by deterministic models from a theoretical perspective [16], as it may result in undesirable properties, e.g. the resulting geodesics going outside of the data manifold.

We therefore consider using the approximate curve energy / distance of the straight line (in the Euclidean sense) connecting the representations in the latent space:

$$RR^{geo}(z; \mathcal{A}_{\mathcal{X}}) = \bigoplus_{a_i \in \mathcal{A}_{\mathcal{X}}} \mathcal{E}(\tilde{\gamma}(z, E_{\mathcal{X}}(a_i)))$$

where $\tilde{\gamma}(z_1, bz_2) = (1-\alpha)z_1 + \alpha z_2$ is the convex combination between the points $z_1, z_2$. Further descriptions on our method for obtaining the geometric representations can be found in Section A.1.

## 3 Experiments

In the following we will evaluate relative geodesic representations on the latent communication problem across models trained with different initializations, different architectures, and tasks.

### 3.1 Aligning neural representational spaces trained independently

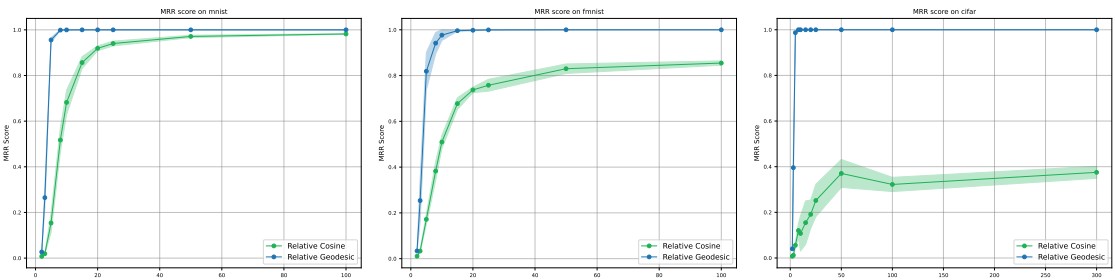

Figure 1: *Aligning latent spaces of autoencoders*: MRR score as a function of the number of anchors on pairs of autoencoders trained with different initializations on the MNIST (left), FashionMNIST (center), CIFAR10 (right) datasets respectively. In green, we plot the performance of [32], in blue, our method. Shaded area indicates standard deviation across different random set of anchors. Relative geodesics consistently outperform the cosine baseline, obtaining peak performance.

**Experimental setting** For the following experiment we trained pairs of convolutional autoencoders $(F_1, F_2)$ with different initializations on the MNIST [11], FashionMNIST [37], CIFAR10[23] datasets. The architecture of the convolutional autoencoder is detailed in the Appendix. After training we extracted 10k samples from the test set, and map them to the latent spaces of the two models, to representations $\mathbf{Z}_1 = E_1(\mathbf{X}), \mathbf{Z}_2 = E_2(\mathbf{X})$ respectively. Starting from a small set of anchors in correspondence $\mathcal{A}_{\mathcal{X}} \mapsto \mathcal{A}_{\mathcal{Y}}$, the objective is to evaluate how well from the relative representations is possible to recover the full correspondence between the representations $\mathbf{Z}_1, \mathbf{Z}_2$. As baseline we compare with relative representations using cosine similarity [32].

**Analysis of results** In Figure 1 we plot the performance in terms of MRR on MNIST,FashionMNIST, CIFAR10 datasets. To obtain the score we first compute similarity matrices between relative representations of the two spaces as $\mathbf{D}(\mathbf{Z}_1, \mathbf{Z}_2)$ where $\mathbf{D}_{i,j} = \frac{RR(\mathbf{Z}_1)_i^T RR(\mathbf{Z}_2)_j}{\|RR(\mathbf{Z}_1)_i\|_2 \|RR(\mathbf{Z}_2)_j\|_2}$. Then we compute the Mean Reciprocal Rank (MRR, see Appendix A.2.1) on top of the similarity matrix. In the figure we plot MRR as a function of a random set of anchors, where the shaded areas indicate the standard deviations over 5 different set of random anchors with the same cardinality. Our method consistently performs better than Relative Representation, saturating the score with few anchors on all the domains, despite the different degree of complexity of the latent spaces. In addition, our method show way less variance in the result, being more robust to the choice of the anchor set.

**Takeaway**: Relative geodesic representation capture almost perfectly the transformations between representational spaces of models initialized differently, outperforming [32] in terms of number of anchors needed and robustness.

### 3.2 Stitching autoencoder models

**Experimental setting** For this experiment we consider the same pairs of autoencoders trained on the MNIST, FashionMNIST, CIFAR10 datasets of section 3.1. Starting from a set of five random anchors we want to

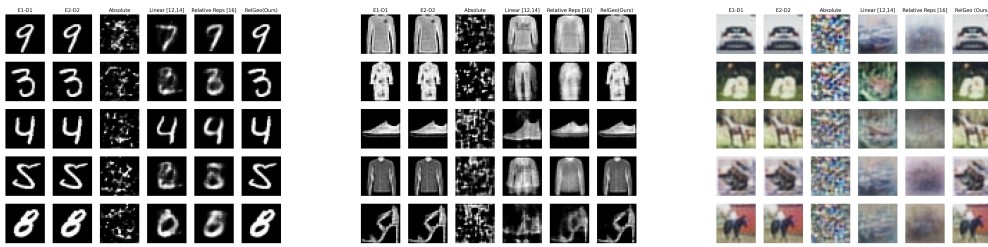

Figure 2: *Stitching on Autoencoders*: We visualize qualitative reconstructions of samples, stitching autoencoders of models trained with different initializations on `MNIST` (left), `FashionMNIST` (center), `CIFAR10` (right). The first two column shows reconstructions from the original models; middle columns represent baselines [27, 25, 32]; the rightmost column is our method. Relative geodesics yield the best stitching results using just 5 anchors.

estimate a transformation $T$ between the model representational spaces $Z_1, Z_2$. Differently from [32], in which zero shot stitching was achieved by training once a decoder module with relative representations and then exchanging different encoder modules, here we achieve stitching without training any decoder. We compute relative representation with respect to the set of anchors, and then compute a similarity matrix $\mathbf{D}(\mathbf{Z}_1, \mathbf{Z}_2)$. Then we compute the vector $\mathbf{c} = \arg\max_i(\mathbf{D})$ representing a correspondence between the two representations matrices $\mathbf{Z}_1, \mathbf{Z}_2$, and use $c$ to fit a linear transformation $T$ to approximate the transformation between the two domains. We perform stitching by performing the following operation for a sample $x \in \mathcal{X}$: $\tilde{x} = D_2 \circ T \circ E_1(x)$.

**Analysis of results** We visualize the results of reconstructions of random samples in Figure 2, comparing with [32, 25, 27]. For each dataset, each column represents respectively: (i) the original autoencoding mapping for a sample $x$ of model $F_1$, $D_1(E_1(x))$ (ii) $D_2(E_2(x))$ (iii) the mapping $D_2(E_1(x))$ (iv) the mapping $D_2(T_{anchors}E_1(x))$ where $T_{anchors}$ is estimated on the five available anchors, (v) the mapping $D_2(T_{cosine}E_1(x))$ where $T_{cosine}$ is estimated among all 10k samples with the correspondence $c$ obtaining in the relative space of [32] (vi) Our result $D_2(T_{relgeo}E_1(x))$ where $T_{relgeo}$ is estimated from the correspondence obtained in the relative geodesic space. While the baselines do not reach a good enough reconstruction quality, reconstructions with our method are almost perfect in accordance with the results in Figure 1.

**Takeaway**: The relative geodesic space enables to stitch together neural modules trained on different seeds.

### 3.3 Zero shot Stitching of vision foundation models

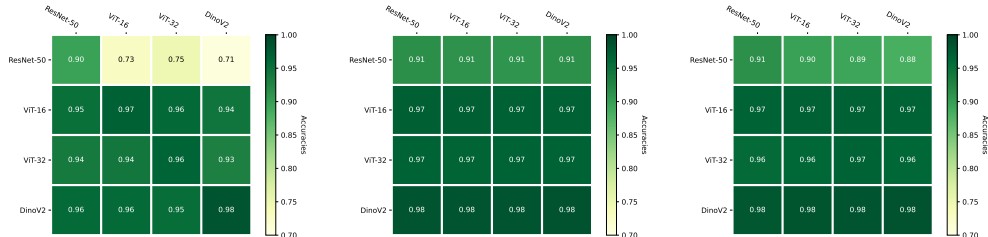

Figure 3: *Stitching of vision foundation models*: We visualize the accuracies of models stitched together on a classification task on `CIFAR10`. We plot `cos`[32] (left), `geo2` (center), `geo1` (right) representations. `geo1` results in accuracies that are not significantly degraded even when performing model stitching.

**Experimental setting** We perform experiments on pretrained classifiers from Hugging Face, investigating the accuracies of stitching together different backbones with classfication heads, on `CIFAR10 dataset`. For this experiment we follow the stitching procedure of [32], section 5. We consider ResNet-50 [18], Vision Transformers (ViT) [13], with both patch 16-224 and patch 32-384, and DinoV2 [33]. These models differ in architecture and pretraining tasks (classification, self supervised contrastive learning). We mainly compare cosine relative representation (`cos`)[32], relative geodesic representation of the curve length assuming Euclidean geometry on the logits with 20 discretization steps of Equation 1 (`geo1`) and directly using the distance of the corresponding logits (`geo2`) as relative representations, corresponding to 1 discretization step.

**Analysis of results** The accuracies are shown in Figure 3. We plot confusion matrices of accuracies indicating that the performance of stitching the backbone of model on each with the classification head of each column. The accuracies are shown in Figure 3, while the MRR with respect to cosine similarity are shown in Figure 4.

**Takeaway:** Using geometric relative representations yields better accuracies, avoiding downgrading of performance when performing model stitching.

## 4 Conclusions

In this work we explored the framework of relative representation equipped with geodesic energy to capture the trasformations occuring between neural manifold learn by distinct neural architecture. We demonstrated superior performance. As limitation we observe that the evaluation results depend on the number of discretization steps when evaluating the representations. Future steps include exploring the multimodal case, when $\mathcal{X} \neq \mathcal{Y}$, different formulation of the energy, and considering different architectures e.g. VAEs as in [34, 3, 2].

## Acknowledgments and Disclosure of Funding

We thank Gregor Krzmanc, German Magai, Vital Fernandez for insightful discussions in the early stages of the project. HY is supported by the Research Council of Finland Flagship programme: Finnish Center for Artificial Intelligence FCAI and additionally by the the grant 345811. HY wishes to acknowledge CSC - IT Center for Science, Finland, for computational resources. MF is supported by the MSCA IST-Bridge fellowship which has received funding from the European Union's Horizon 2020 research and innovation program under the Marie Skłodowska-Curie grant agreement No 101034413.

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

## A Appendix

### A.1 Obtaining geometric representations

We use the curve energy / length of a curve to form the relative representations. One can consider two cases, using Euclidean geometry on the logits and using the Fisher Information Matrix induced by the output probabilities.

The energy / length of a curve is given by the following [34], where $v(t) = \dot{\gamma}(t)$

$$\mathcal{E}(\gamma) = \frac{1}{2} \int_a^b v(t)^\top G(t) v(t)^\top \mathrm{d}t, \tag{2}$$

$$d(\gamma) = \int_a^b \sqrt{v(t)^\top G(t) v(t)^\top} \mathrm{d}t. \tag{3}$$

The energy / length can be approximated using discretizations as follow

$$\mathcal{E}(\gamma) = \sum_{i=1}^N E_i = \frac{1}{2} \sum_{i=1}^N v(t_i)^\top G(t_i) v(t_i) \Delta t,$$

$$d(\gamma) = \sum_{i=1}^N d_i = \sum_{i=1}^N \sqrt{v(t_i)^\top G(t_i) v(t_i)} \Delta t,$$

where $\Delta t = \frac{1}{N}$, with $N$ being the number of discretization steps.

When the step size is small enough, the geodesic energy / length on the latent space can be approximated by the geodesic energy / length on the output space [34]. For Euclidean geometry, the geodesic energy / length is clearly given in closed-form as the geodesics are straight lines. For certain Fisher-Rao geometries, e.g. the one induced by categorical distributions, one can derive closed-form expressions of the (approximate) geodesic energy / length of a curve [2, 29].

### A.2 Additional details

#### A.2.1 Mean Reciprocal Rank

Mean Reciprocal Rank (MRR) is a commonly used metric to evaluate the performance of retrieval systems [32]. It measures the effectiveness of a system by calculating the rank of the first relevant item in the search results for each query.

To compute MRR, we consider the following steps:

1. For each query, rank the list of retrieved items based on their relevance to the query.

2. Determine the rank position of the first relevant item in the list. If the first relevant item for query $i$ is found at rank position $r_i$, then the reciprocal rank for that query is $\frac{1}{r_i}$.

3. Calculate the mean of the reciprocal ranks over all queries. If there are $Q$ queries, the MRR is given by:

$$\text{MRR} = \frac{1}{Q} \sum_{i=1}^{Q} \frac{1}{r_i} \tag{4}$$

Here, $r_i$ is the rank position of the first relevant item for the $i$-th query. If a query has no relevant items in the retrieved list, its reciprocal rank is considered to be zero.

MRR provides a single metric that reflects the average performance of the retrieval system, with higher MRR values indicating better performance.

### A.2.2  Architectural details

We provide here the architectural details of the convolutional Autoencoders employed in experiments in Figures 1 and 2

| Encoder |
| --- |
| $3 \times 3$ conv. 32 stride 2-ReLu |
| $3 \times 3$ conv. 64 stride 2-ReLu |
| Flatten |
| $(64 * k * k) \times h$ Linear |
| Latents |
| **Decoder** |
| $h \times (64 * k * k)$ Linear |
| Unflatten |
| $3 \times 3$ conv. 64 stride 2-ReLu |
| $3 \times 3$ conv. 32 stride 2-ReLu |
| Sigmoid |

Table 1

For the classifier experiment, in order to obtain geometric representations we need a decoder. The architecture is shown in Table 2.

| Classification head |
| --- |
| $input\_dim$ LayerNorm |
| $input\_dim \times 500$ Linear-Tanh |
| $500 \times 10$ Linear |

Table 2

For evaluating the performances of the representations, we train a classifier with the same architecture as used by [32].

### A.3  Related Works

**Representation alignment:**   There is a growing evidence that neural networks trained under different settings still tend to generate similar internal representations [7, 22, 20, 26, 6, 28, 19], which is shown to be more evident in wide and large networks [5, 31, 35]. These aligned representations make it possible to stitch models together [14, 4, 10], allowing the swapping of components between different networks.

**Latent geometries** [34, 36] considered the latent space of autoencoders, proposing to use a pullback metric, assuming the output space is Euclidean. For classifiers one can obtain a Riemannian metric primarily using two approaches [15], either by pulling back the Fisher Information Matrix [1, 2] or by assuming an Euclidean geometry on logit space and pulling back the metric.

## A.4 Additional results

We show the MRR results of the representations on real models in Figure 4. Surprisingly, using relative geodesic representations results in loss of MRR.

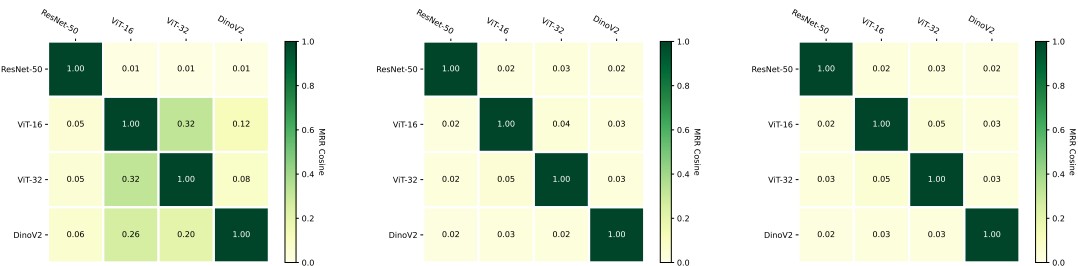

Figure 4: *MRR of classifiers*: `cos` (left), `geo1` (center), `geo2` (right)

