# OpenReview forum: "Connecting Neural Models Latent Geometries with Relative Geodesic Representations"
_NeurIPS.cc/2024/Workshop/UniReps — UniReps_

### Official Review · Reviewer_N8LJ · 2024-10-01
**Promising novel development in relative representations**

**Rating:** 8
**Confidence:** 4

**Review:**

The paper is well written, easy to follow and understand. The introduction into the framework and the mathematical background is very clear and well motivated, the use of geodesics for relative representations seems convincing. While relative representations and all experiments presented are not novel, the use of geodesics provides a novel approach to relative representations which improves those quite significantly. The experiments are relevant and well-motivated, however the discussion of results is lacking and should be extended/included. The take-away message after each result is a very nice touch!

**Questions:**

3.1
- How are anchors chosen?
- Are the different models just initialized with a different random seed? Do all models achieve the same performance?
- Why do you use MRR, which kind of information does this measure and what does it show wrt relative representations?
- Can you also include a linear baseline?

3.2
- Do you only use 5 anchors? How do the results look with more anchors (RR based on cos seem to need more anchors based on Fig.1)?

3.3
- Is geo1 or geo2 better, you mention geo1 in the figure cpation but based on the figure geo2 (center) seems to perform slightly better?
- What is geo2 exactly? Is this also based on geodesics?
- What is the take-away from the MRR results?

**Suggestions for improvement:**
- Increase font size on all figures and include titles for Fig.3 to increase readability
- Use different color scale for Fig.3 to make it easier to spot differences between methods
- Describe the actual results in “analysis of results” and not the what you plotted, description of the plot should be moved to figure captions and the analysis should not be referred to the reader
- L.140 spelling: trasformations -> transformations
- Appendix: include captions for tables and expand on A.4

This work is of quite high significance and interest for the community and I recommend acceptance.

---

### Official Review · Reviewer_rvvH · 2024-10-03
**Using Geodesics for Relative Representations**

**Rating:** 7
**Confidence:** 4

**Review:**

**Summary:**

The work uses the pull-back metric from the output space as a distance metric for relative representations.
They show that this better preserves accuracy when stitching together CIFAR-10 classifiers compared to using cosine similarity,
and also that this allows for stiching of latent spaces with a linear transformation.


**Questions and Suggestions:**

Q1: If you have decided to call your kind of relative representations "relative geodesics", then I suggest introducing the name in (i) in the introduction, e.g.
"(i) We propose a new representation that capture the isometric transformation between data manifolds learned by distinct models, called relative geodesics."
This is to avoid confusion when you use the term later.

Q2: On line 77 and 78 it says: "We therefore consider using the approximate curve energy / distance of the straight line (in the Euclidean sense)".
I assume you mean "length" of the straight line. I also assume that what you mean is that you approximate curve lengths using straight lines on the output space,
then pull it back to the latent space and use this as the distance measure. However, I find this is very unclear in the text. Especially since you often write
energy/length in the citation above and in all of A.1. Writing it in this way causes confusion, since the energy and length of the curve are two different
things. So when you write you use the energy/length, it sounds like you use both. Please clarify.

Q3: There is a typo in the equation on line 79: \gamma(z_1, bz_2) should be \gamma(z_1, z_2).

Q4: The procedure in experiment in 3.1 is unclear. It says: "Starting from a small set of anchors in correspondence A_X -> A_Y, the objective is to evaluate how well from the relative
representations is possible to recover the full correspondence between the representations". What do you mean by "recover"?
Since you are using the Mean Reciprocal Rank (MRR), you must be using something as queries. Since you write (94-95) "Then we compute the Mean Reciprocal Rank
(MRR, see Appendix A.2.1) on top of the similarity matrix." It sounds like the similarity matrix is the query? What do you retrieve?
Please also clarify what you use as relevance measure to rank the items and how you decide which is the "first relevant item in the list".

Q5: Please write the full name the first time you introduce a new concept. I suggest writing the full name of Mean Reciprocal Rank in both the
caption of figure 1 and in line 92. You can then use the abbriviation in line 94.

Q6: In line 97, it says: "Our method consistently performs better than Relative Representation...", you mean "Using curve energy consistently
performs better than using cosine similarity". They are both relative representations.

Q7: In A.4 MRR seems to be quite low for both curve energy and cosine similarity. Do you have any idea why?
Especially why it is so different from the experiment in 3.1? Please also add details about how you calculated the MRR in this case.


**Strengths:**

S1: I like the idea of using geodesics for the distance measure

S2: Using geodesics for the relative representations, they show that the representations can then be connected using a linear transformation.

S3: They average over multiple choices of anchors in 3.1 to account for the variability incurred by random choice of anchors.

S4: The structure in section 3 with "Experimental setting", "Analysis of results" and "Takeaway" is very nice.


**Weaknesses:**

W1: The discription of how they construct the distance measure is unclear, see Q2.

W2: The description of their experiment 3.1 is unclear, see Q4.


**Justification:**

I recommend this paper to be accepted to the workshop with the added clarifications.

The idea is good even though some integral parts need to be rewritten.

---

### Official Review · Reviewer_1Mjr · 2024-10-06
**Conditions under which the results hold unclear**

**Rating:** 4
**Confidence:** 4

**Review:**

### **Paper summary**

The paper proposes comparing different models’ latent representations (relative representations, RR) according to Riemannian geodesic distances, presumably a more natural metric compared to cosine similarity that has mostly been prevalent so far. Due to intractability, the paper resorts to using the Euclidean distance. The paper empirically finds superior RR, both in terms of a MRR metric and autoencoder reconstructions based on stitching RR across models.

### **Strengths**

It is important to point out possible limitations in using cosine similarity as a metric, and proposing Riemannian geometry as a natural alternative, even if it ultimately is not used in the paper. The empirical findings on improvements are surprising and potentially valuable if persistent.

### **Weaknesses, questions and suggestions**

- Why is MRR used? What are alternatives that have been rejected?
- Cosine similarity a priori is a natural metric for neural network’s latent spaces, as latent representations tend to be linear, because they will be processed by a linear readout at the end of the network. However, it discards information about the norm of the vector. Would the authors expect a dot-product metric $z\_1 \\cdot z\_2$ that incorporates both angle and norm to be superior to Euclidean distance?
- Would the same results have been achieved if one just had proposed using Euclidean distance $|z\_1 \- z\_2|$ instead of $\text{cosine  similarity}(z\_1, z\_2)$? I.e., is the Riemannian framework strictly needed for the paper’s empirical results?
- It is unclear to me in how far the results on superiority are contingent on data preprocessing. E.g., is it possible to state whether superiority persist if data is centered and normalized? Making code available would be helpful to this end.
- Limitations are not discussed.

### **Recommendation**

The core weakness is to fully comprehend the processing conducted in the paper, which is why I right now recommend rejection.

---

### Official Review · Reviewer_uVT2 · 2024-10-06
**incremental improvement to relative representations alignment with impressive results**

**Rating:** 9
**Confidence:** 4

**Review:**

Authors propose a new geodesic representation that leverages the differential geometrical structure of latent spaces to model the transformations between them, demonstrating improved performance compared to previous methods. This approach is shown to be effective with experiments with aligning and stitching models, and the results are impressive. I did not completely follow the derivation of the the curve energy used to form the relative representations and would like to have seen some discussion of the runtime complexity of the implementation used. But I am familiar with relative representation research and found the background well covered and experiments convincing.

Rating provides is relative to the abstracts track as this appears to be an excellent fit - providing a short and insightful finding very relevant to the workshop audience.

---

### Decision · Program_Chairs · 2024-10-10

**Decision:**

Accept

**Comment:**

In light of the positive reviewers' feedback and relevancy of the submission, we are pleased to accept this paper for presentation at UniReps 2024. We kindly ask the authors to incorporate the reviewers' suggestions and feedback in the final camera-ready version of the manuscript.